

# Web-based macroseismic intensity study in Turkey: entries in Ekşi Sözlük

Deniz Ertuncay[1], Laura Cataldi[1], and Giovanni Costa[1]

[1]University of Trieste, Department of Mathematics and Geosciences, via E.Weiss 4, 34128 Trieste, Italy

**Correspondence:** Deniz Ertuncay (dertuncay@units.it)

**Abstract.** Ekşi Sözlük is one of the most visited websites in Turkey. Registered users of the website share their knowledge about any topic on an internet forum-like environment. In this study, we collect the user entries on the topics of 20 earthquakes in Turkey and the surrounding area. Entries with city and district level information are converted to intensity values. Shape maps of the earthquakes are created by using a ground motion to intensity conversion equation. User entries and created shake maps are compared. It is found that entries correlate with the predicted intensities. It is also found that local soil conditions and building types have an amplifier effect on entries in the web site. Several entries in the earthquake topics have magnitude estimations. Difference between predicted and observed intensities also vary with distance. Users are able to predict the magnitudes of the earthquakes with $\pm$ 0.54 misfit. This study shows that Ekşi Sözlük has a potential to be a reliable source of macroseismic intensity for the earthquakes in Turkey, if the felt reports have been collected with a predetermined format.

## 1 Introduction

Many national seismic data providers (British Geological Survey, 2020; Swiss Seismological Service, 2020; Zentralanstalt für Meteorologie und Geodynamik — ZAMG, 2020; Agencija Republike Slovenije Za Okolje - ARSO, 2020; Sbarra et al., 2010) along with international organizations (European-Mediterranean Seismological Centre (EMSC) Bossu et al. (2017), United States Geological Survey (USGS) Wald et al. (2012)) collect web-based macroseismic survey data. Various questions are asked in form of a questionnaire to the individuals who are willing to share their experience after an earthquake. The answers are then converted to macroseismic intensity scales, and felt maps are created as an end product of the earthquake. The data may be collected continuously as in EMSC and USGS as well as after a specific earthquake (Bossu et al., 2015; Liang et al., 2017; Bossu et al., 2008).

In Turkey, Disaster and Emergency Management Presidency of Turkey (AFAD) is the only data collector for macroseismic intensity. However, felt reports of AFAD are rare and lack location information. On the other hand, Turkish data providers were the 7th largest data providers (3rd in Europe) to EMSC in 2018.

Internet users in Turkey are also sharing their experiences on earthquake in other websites such as Ekşi Sözlük (www.eksisozluk.com). Ekşi Sözlük is a collaborative dictionary. To be a full member of the website, newly registered users are required to write 10 entries in existing topics. Then the entries are analyzed according to the rules of the site. By doing that Ekşi Sözlük provides relatively reliable information to the visitors. It was the 14th most visited website in Turkey in 2019

(Alexa Internet, 2020). Even though there are entries with other languages, the main language used on the website is Turkish. Users of the website create topics for earthquakes that they have felt and also for major earthquakes occurring around the world.

In this study, we have collected entries from various major earthquakes in western Turkey. Entries which provide city

and district information are analyzed. Macroseismic intensity maps are created for the earthquakes by using the magnitude information. Analyzed entries are compared with predicted intensities. Furthermore we collect the magnitude guesses of the users and compare them with the real results.

## 2 Data

To analyze the entries in the earthquake topics, 20 titles with the highest number of entries for earthquakes that have occurred

in western Turkey and the surrounding area are selected (Table 1). Western Turkey is chosen due to higher access to the internet (Turkish Statistical Institute, 2019). Entries that provide location information in district levels are filtered. These entries are labelled according to the modified Mercalli-Cancani-Sieberg (MCS) scale (Sieberg, 1930). Several entries also provide a guess of the magnitude of the earthquake.

Topics are created almost immediately after the arrival of S waves. Several topics are created for some earthquakes that are

widely felt in many regions. Moderators of the website combine the topics into the proper topic title which is in general the date of the earthquake and the nearest major city or the name of the sea to the epicenter. Topics are created with the same format, which is provided below:

10-aralik-2019-balikesir-depremi–6277350

day-month-year-location-earthquake–topicID

When an aftershock is felt, the entries indicate the existence of an aftershock. In these cases, we subdivided the earthquake

to multiple earthquakes by adding a number to the end of the topicID. Several MCS labels, along with the entry, can be seen below:





Topic: 20-aralik-2018-yalova-depremi–5881852

Entry: "I felt it in Gebze (Kocaeli) but I was the only one.

Lights are not swinging, none of my family members have felt it."

MCS:1

Topic: 20-subat-2019-canakkale-depremi–5947040

Entry: "I felt it strongly. I think it was M4.4 ...

MCS:2

Topic: 30-kasim-2018-yalova-depremi–5861331

Entry: "I thought my neighbor downstairs hit the ceiling of their

apartment with a hammer ... "

MCS:3

Topic: 26-eylul-2019-istanbul-depremi–6191375

Entry: "We saw the rattling windows not only in our office building

but also on the neighboring building. I tried to walk forward but instead

I staggered backwards ... "

MCS:4

Topic: 6-subat-2017-canakkale-depremi–5296414

Entry: "We felt it stronly. We evacuated the office building."

MCS:5

Topic: 26-eylul-2019-istanbul-depremi–6191375

Entry: "I was at Avcilar. I saw falling parts from a building ... "

MCS:6

Hereafter, earthquakes are represented by their topicID. In Table 2, we present information about the entries. 5392358 is excluded from further analysis due to lack of entries.

## 3 Method

To calculate the intensity values for a given event, peak ground velocity (PGV) and peak ground acceleration (PGA) values were calculated using ground motion prediction equations (GMPEs) and then converted to intensity values. The methodology is taken from Cataldi et al. (in prep.), where they use 90 ($M_l > 3.4$) events occurred in Italy between 1972 and 2016. In





particular, PGV and PGA values are extracted from the seismograms (Gallo et al., 2014) and correlated with intensity values

coming from the Italian Macroseismic Database (DBMI15). All intensity points are binned in integer (1.0) intervals. In addition

to the standard procedure of fitting a linear regression to derive ground motion to intensity conversion equations (GMICE), an

alternative approach is also given in the form of Gaussian Naïve Bayes classifiers (GNB). The GNB methodology does not

provide explicit equations and a learning algorithm is used instead; more details can be found in (Cataldi et al., in prep.). In this

study, GNB results are used to build a piece-wise GMICE function relating peak ground motion values to intensity. Ground

motion values are checked for the interval in which they fall and are assigned to the corresponding intensity (I) value. For PGA

and PGV, the used intervals are the following:

The PGA-based function is used for events with $M_w < 5$, the PGV-based one is used for events with $M_w > 5$.

## 4    Results

All topics are analyzed with the methods explained in Section 2 and Section 3. 4 of them are represented in this section. Maps

of all the earthquakes can be found at Github repository of the study. We compare the computed intensities with the entries;

and we also compare them with the EMSC felt report maps. Furthermore, magnitude guesses are compared with the measured

magnitudes.

### 4.1    12th June 2017 Lesbos Earthquake

12th June 2017 Lesbos earthquake, $M_w = 6.2$, (topicID: 5388936) had occurred south of Lesbos island and felt largely in

Turkey. Izmir is the closest city to epicenter of the earthquake, thus topic is created as 12th June 2017 Izmir earthquake in Ekşi

Sözlük. There are 151 of entries with city and district information. Farthest felt entry is from Pursaklar district (≈580 km) of

Ankara (MCS = 1). Entry with largest MCSs are from Konak (≈80 km), Karaburun districts (27 km) of Izmir and Yunusemre

district (≈85 km) of Manisa (Fig. 1). In Karaburun district, user overbalanced during the earthquake with many others inside

the government office. Cracks on walls and columns of the buildings are reported in two entries. Intensity map has strong

correlation with the labelled maximum MCS scale. Minimum MCSs are hard to interpret since some of the entries contain

only words such as "strongly felt", which is labelled as MCS = 2. Such entries lowered the average MCSs. Hence we preferred

to rely on maximum MCS labels.

A large amount of entries are written in Istanbul (Fig. 2). Almost all districts of the city have maximum MCS of 2 except

Avcilar district which has 3. This can be linked with the loose soil of Avcilar district of Istanbul. Avcilar district was affected by

the $M_w = 7.4$ Izmit earthquake of 1999. More than $10\,\%$ of the buildings were either destroyed or damaged in the earthquake

(Tezcan et al., 2002). It is due to the amplified shaking and soft sediments in the district (Tezcan et al., 2002; Ergin et al., 2004;

Akarvardar et al., 2009).

There are 635 felt reports at AFAD for the earthquake. It is hard to interpret the data due to lack of location information

of the earthquake. There are intensity values of 11 in several reports which is highly unlikely for magnitude 6.2 earthquake.

EMSC have 755 felt reports for the earthquake (Fig. 3). Intensities in EMSC are larger than the ones in Ekşi Sözlük. There are





intensity 5 reports in far away cities such as Istanbul and Sofia which is unlikely. Furthermore, there are various intensity 10+ in EMSC in Izmir. However, there is no report that supports such destruction in Izmir. Various masonry buildings have cracks on their walls but none of them have collapsed (9 Eylul, 2017).

### 4.2   25th August 2019 Ankara Earthquake

25th August 2018 Ankara earthquake, $M_l = 3.5$, (topicID: 6155192) had occurred south of Kecioren district of Ankara and was felt locally in the city (Fig. 4). There are 129 entries with city and district information. Depth of the earthquake is measured as $5\,\mathrm{km}$. Even though there are large number of magnitude 3.5 earthquakes in Turkey, this was felt by many inhabitants since hypocenter was located beneath the city of Ankara and the earthquake had a shallow hypocentral depth. Due to the shallow depth, almost all districts provided relatively higher maximum MCS values.

There is no felt report in AFAD for the earthquake. There are 205 felt reports in EMSC (Fig. 5). Intensity measures of Ekşi Sözlük and EMSC are highly correlated for the earthquake.

### 4.3   26th September 2019 Istanbul Earthquake

26th August 2019 Istanbul earthquake, $M_w = 5.6$, (topicID: 6191375) had occurred south of Silivri district of Istanbul and was felt largely in Istanbul and the surrounding cities (Fig. 4). There are 233 entries with city and district information. A minaret of

a mosque in Avcilar district of Istanbul was collapsed (Hurriyet, 2019) and more than 450 buildings were damaged (Anadolu Agency, 2019). The earthquake provided the largest dataset with 233 comments with district level information on 12 cities. It is due to the fact that Istanbul is the most crowded city in Turkey and the earthquake happened in the daytime (13:59 local time).

In Silivri, Buyukcekmece and Avcilar districts, maximum MCS from the entries are labelled as 6. Intensity map predicts the

MCS 4.5 for Avcilar district. Local soil conditions as explained in Section 4.1 may have a role on the exaggerated intensities in entries written in Avcilar.

In Besiktas district, maximum MCS of 5 is given in 3 entries. All of them are due to the evacuation of the buildings. Two of these entries are from high-rise office buildings, which probably caused extra panic due to the swing of the tall buildings. The evacuation also influenced by panic is also one of the reasons for the maximum MCS of 5 that is given to the entries from

Fatih, Beyoglu and Kadikoy districts. In Kartal district, MCS = 5 is due to the fallen objects from shelves.

There are 70 felt reports in AFAD for the earthquake with maximum intensity of 6, which is also the case in Ekşi Sözlük. In EMSC (Fig. 7) there are 2027 felt reports. In Istanbul, intensities are reported slightly higher with respect to Ekşi Sözlük. However, it is important to keep in mind that felt report in EMSC is designed for this purpose, whereas in Ekşi Sözlük, entries are written in free format. In general, EMSC and Ekşi Sözlük are correlated in Buyukcekmece and Avcilar districts. On the

other hand, intensities are at least one grade lower in Ekşi Sözlük with respect to EMSC. There are large intensity values in cities such as Eskisehir ($d > 200\,\mathrm{km}$) and Denizli ($d > 350\,\mathrm{km}$). Large intensities in such far away distances for a $M_w = 5.6$ earthquake are unlikely.





### 4.4 10th October 2019 Yalova Earthquake

10th August 2019 Yalova earthquake, $M_w = 4.0$, (topicID: 6208576) had occurred north of Yalova and was felt in Istanbul
and the surrounding cities (Fig. 8). There are 68 entries with city and district information. Most of the entries are coming from
Kartal district of Istanbul and Gebze district of Kocaeli. In Gebze, maximum MCS is 5, which is due to the evacuation of a
building with panic. The rest of the entries in Gebze are claiming 2 and 3. MCS scale is highly subjective on an individual's
feelings on earthquakes. Thus, users in Ekşi Sözlük are writing their feelings without using any guidelines. As in this example,
unexpectedly high MCSs may occur.

There is no felt report in AFAD for the earthquake. There are 1371 felt reports in EMSC (Fig. 9). Intensity measures of Ekşi
Sözlük and EMSC are correlated in Gebze, Kartal, Kadikoy and Kucukcekmece districts. There are more datapoints in EMSC
than in Ekşi Sözlük, which provides more information in different regions of the area, especially in the city of Yalova. There
are several unexpectedly high intensity values for Istanbul in EMSC.

### 4.5 Magnitude Guesses

Users in Ekşi Sözlük also provide magnitude information depending on their feelings. We include entries without location
information. In various entries, the magnitude is guessed with a semi-infinite range, such as "...it was at least 4.8" in 20th
February 2019 Dardanelles earthquake (topicID: 5947040), or it is guessed within a full range, such as "...it is between 2-3
..." in 10th December 2019 Balikesir earthquake (topicID: 6277350). In semi-definite guesses, given edge value is considered.
The average of the range is used when it is provided.

Measured magnitudes along with the user guesses can be seen in Fig. 10. The average misfit between guessed magnitudes
and measured magnitudes is 0.54. We also examine the earthquakes with at least two guesses. Misfit is calculated, again, 0.54.
However, it is important to keep in mind that, guesses with range are averaged and various guesses are semi-definite.

## 5 Discussion and Conclusion

In this study, we gather entries from earthquake topics in Ekşi Sözlük. In the topics, users discuss the earthquake and their
experiences. We filter the entries with city and district level information that can be converted to MCS scale. When there is an
aftershock, it is discussed in the same topic on the website. In such incidences, we divide the topic into sub topics.

In total 27 earthquakes are chosen for the analysis. Intensity maps are created for the earthquakes and correlation between the
predicted intensities and the entries which were converted to intensity values are roughly interpreted. The values are mapped to
the districts of Turkey due to lack of precise location information. Interpretation is done over maximum MCSs. Uncertainty of
the data points in terms of location varies depending on the epicenter of the earthquake and the positioning of the governmental
district.

To have an insight of the relation between the predicted MCS and observed MCS, a rough relation between the two param-
eters and the distance is analysed. Observation points are labelled with their districts. However, the district is most likely to





have more than one intensity inside its border depending on the epicentral distance from the earthquake and the border of the
district (Gebze district in Fig. 8 has 3 different MCS values inside its border). To overcome this problem, we have calculated
the centroid point of each district and treated the district as a point.

We have calculated the residuals of MSC differences between the predicted and observed values (Fig 11). To do that, we
binned the distance between the epicenter and the centroid points with 10 km intervals from 0 km-100 km. We combined more
distant points with +100 km label. Weighted average of average MSCs are used for the residual calculations. Asymmetric errors
are calculated by using the minimum and maximum MSC value for each bin and the predicted MSC. If all data points have the
same MSCs, then errors are not calculated. A line is fitted to residuals when more than one data is associated in different bins.

It is found that residuals tend to increase with increasing distance. In longer distances, MSC values are more likely to be 1.
When there is an entry which states the feeling of the earthquake, it is more likely to have MSC value of 2. This is due to the
fact that it is hard to distinguish between MCS 1 to MCS 2 by analyzing the entries. The website is not dedicated to provide
exact information on earthquakes.

Furthermore, population distribution of districts are mostly heterogeneous. An example can be seen in Section 4.3. Silivri
district of Istanbul has all of its EMSC data from the coastline. However, its centroid position is located in northern part of the
district, for which MSC value is one integer lower with respect to the highly populated coastal area.

In 10th October 2019 Yalova Earthquake, aftershocks (6208576-2, 6208576-3, 6208576-4) mostly have 1 of residuals. City
of Istanbul and Kocaeli have many reports with MSC = 2. However, all the centroid points of the districts is in MSC = 1 area.

Despite not having the exact location and subjective feelings of the users, it is found that the district with loose soils have
relatively bigger maximum MCS values. Avcilar district in Istanbul has the highest MCS value (MCS = 3) for the Lesbos
earthquake (Section 4.1) in the city. In the Istanbul earthquake (Section 4.3), Avcilar also has the highest MCS (MCS = 6) with
respect to other districts with the same epicentral distance. Avcilar district has suffered during 1999 Izmit earthquake due to
the amplified shaking and soft sediments in the district (Tezcan et al., 2002; Ergin et al., 2004; Akarvardar et al., 2009).

Entries that are written from high-rise buildings were affected more due to these buildings' tendency to amplify the motions
of longer periods (eg. Besiktas district in the Istanbul earthquake). Another reason for higher intensities with respect to the
predicted ones, is the evacuation of buildings, even if it is not necessary. There are unexpectedly high intensity values in
various districts both in Istanbul and Yalova earthquakes.

In 20th August 2019 Ankara earthquake (6148327), there is a large gap between the predicted and observed MSCs. It is due
to the fact that people of city of Ankara are not used to feel an earthquake with respect to seismically active cities such as Izmir,
Istanbul and Yalova. Moreover, the epicenter of the earthquake is close to the city center.

Effect of epicentral depth in small earthquakes can be seen in Ankara earthquake (Section 4.2). Districts of Sincan, Yeni
Mahalle, Mamak and Cankaya have at least one degree higher maximum MCS than the predicted ones.

We also analyzed the magnitude guesses in entries with the measured magnitudes. Users of the website try to guess the
magnitudes, most probably, by comparing their experiences with previous earthquakes. Even though there are good matches in
some of the earthquakes, in most of the cases magnitudes are guessed with 0.54 misfit.

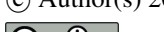

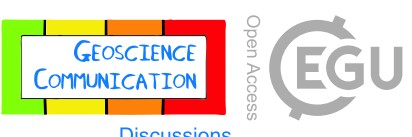

In conclusion, entries in Ekşi Sözlük can provide intensity distribution of earthquakes with limits. Entries are written in free form which creates uncertainties in the MSC labeling process. Entries do not reveal the exact position of the data provider which makes it hard to analyze the differences between observed and predicted MSC values. Despite the limitations, gathered data have similarities with the predictions. The website can provide near real-time intensity information after an earthquake.

## 5.1 Future work

Ekşi Sözlük has potential to provide better information after an earthquake. Fast response time of the users may be useful for having early information about the intensity distribution. A questionnaire can be embedded to the earthquake topics, which would allow us to homogenise the data and analyse it more accurately. The user may fill out a questionnaire along with the subjective entry.

Users may specify further information, such as the details of the building (eg. construction material and year, number of floors etc.) to create more accurate maps or create higher level of information about the earthquake. Level of certainty can be improved by adding location information.

Intensity maps and user feedback about the earthquake can be used for the creation of real time intensity maps and can be published for each topic.

*Code and data availability.*  Python codes that are used to retrieve the user comments, dataset and intensity maps can be found in Github repository of the study.

*Author contributions.*  Deniz Ertuncay analyzed the entries in Ekşi Sözlük website and determined the intensity values. Laura Cataldi created shake maps and Deniz Ertuncay visualized the intensity values collected from Ekşi Sözlük and EMSC to create figures. Results are interpretted by all authors.

*Competing interests.*  The authors declare that they have no conflict of interest.

*Acknowledgements.*  We would like to thank to Enrico Magrin who was a part of SeisRaM working group of Department of Mathematics and Geosciences in University of Trieste (currently works at Istituto Nazionale di Oceanografia e di Geofisica Sperimentale) for his help on the calculation of intensities and Hafize Başak Bayraktar from University of Napoli Federico II department of Structures for Engineering and Architecture for sharing district level polygons of Turkey. We also would like to thank all users of Ekşi Sözlük for sharing their experiences on the website. Finally we would like to thank Selen Caner Ertuncay for proofreading the manuscript.



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





**Figure 1.** Intensity map of 12th June 2017 Lesbos earthquake with labelled entries. Entries are plotted to district level polygons since the exact location of data providers are unknown.





**Figure 2.** Intensity map of 12th June 2017 Lesbos earthquake with labelled entries in West side of Istanbul. Entries are plotted to district level polygons.

**Figure 3.** Felt reports of 12th June 2017 Lesbos earthquake submitted to EMSC on top of the labelled entries.





**Figure 4.** Intensity map of 25th August 2019 Ankara earthquake with labelled entries. Entries are plotted to district level polygons.





**Figure 5.** Felt reports of 25th August 2019 Ankara earthquake submitted to EMSC on top of the labelled entries.

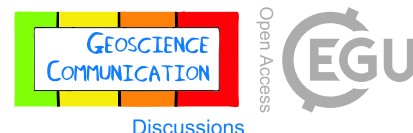

**Figure 6.** Intensity map of 26th September 2019 Istanbul earthquake with labelled entries. Entries are plotted to district level polygons.



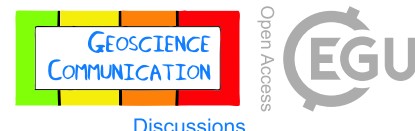

**Figure 7.** Felt reports of 26th September 2019 Istanbul earthquake submitted to EMSC on top of the labelled entries.



**Figure 8.** Intensity map of 10th October 2019 Yalova earthquake with labelled entries. Entries are plotted to district level polygons.





**Figure 9.** Felt reports of 10th October 2019 Yalova earthquake submitted to EMSC on top of the labelled entries.





**Figure 10.** Residuals of measured magnitudes of the earthquakes (Table 1) with average magnitude guesses from users. Earthquakes without any guesses are blank.



**Figure 11.** Residuals of predicted and observed MSCs. Red dashed line represents the baseline. Black circles are the residual of weighted average of the bin. Vertical black lines are the residuals from predicted MSCs and minimum and maximum MSCs that are observed in the bin. Black dotted line is the fitted line to the residual of weighted averages. Number of data points inside bins are provided beneath each bin point with data.





**Table 1.** Topics of various earthquakes in Ekşi Sözlük. First entry, additionally, gives the information of the creation time of the topic. $\Delta t$ is the time between the origin time of the earthquake and the creation of topic of the earthquake. Entries do not have the 'second' information for the time. Magnitudes are moment magnitude unless otherwise stated.

| Topic | Origin Time | First Entry | $\Delta t$ (min) | Magnitude | Event Latitude | Event Longitude | Depth (km) |
|---|---|---|---|---|---|---|---|
| 19-mayis-2011-simav-depremi–2814519 | 19/05/2011 20:15:22 | 19/05/2011 20:15 | <= 1 | 5.7 | 39.14 | 29.1 | 8 |
| 7-haziran-2012-tekirdag-depremi–3418720 | 06/07/2012 20:54:26 | 06/07/2012 20:55 | <= 1 | 5.1 | 40.85 | 27.92 | 14 |
| 8-ocak-2013-ege-denizi-depremi–3673799 | 01/08/2013 16:16:06 | 01/08/2013 16:17 | <= 1 | 5.7 | 39.65 | 25.48 | 8.4 |
| 16-kasim-2015-istanbul-depremi–4966334 | 16/11/2015 15:45:43 | 16/11/2015 15:46 | <= 1 | 3.9 | 40.83 | 28.76 | 7.7 |
| 25-haziran-2016-yalova-depremi–5137587 | 25/06/2016 05:40:11 | 25/06/2016 05:41 | <= 1 | 4.1 | 40.7 | 29.21 | 9 |
| 15-ekim-2016-istanbul-depremi–5208242 | 15/10/2019 08:18:32 | 15/10/2019 08:19 | <= 1 | 4.9 | 42.19 | 30.71 | 10 |
| 6-subat-2017-canakkale-depremi–5296414 | 02/06/2017 03:51:39 | 02/06/2017 03:54 | >1 | 5.3 | 39.56 | 26.02 | 6 |
| 6-subat-2017-canakkale-depremi–5296414-2 | 02/06/2017 10:58:00 | 02/06/2017 10:59 | <= 1 | 5.1 | 39.51 | 26.07 | 6 |
| 12-haziran-2017-izmir-depremi–5388936 | 06/12/2017 12:28:39 | 06/12/2017 12:29 | <= 1 | 6.2 | 38.85 | 26.35 | 10 |
| 17-haziran-2017-izmir-depremi–5392358 | 17/06/2017 03:40:36 | 17/06/2017 03:42 | >1 | 4.6 | 38.91 | 26.22 | 9 |
| 17-haziran-2017-izmir-depremi–5392358-2 | 17/06/2017 19:50:04 | 17/06/2017 19:54 | >1 | 5.2 | 38.85 | 26.44 | 7 |
| 30-kasim-2018-yalova-depremi–5861331 | 30/11/2018 02:36:35 | 30/11/2018 02:37 | <= 1 | 4 | 40.58 | 28.98 | 9 |
| 20-aralik-2018-yalova-depremi–5881852 | 20/12/2018 06:34:25 | 20/12/2018 06:35 | <= 1 | 4.4 | 40.6 | 29.97 | 8 |
| 25-ocak-2019-izmir-depremi–5919561 | 25/01/2019 20:20:33 | 25/01/2019 20:21 | <= 1 | 4.2 | 38.58 | 27.1 | 18 |
| 20-subat-2019-canakkale-depremi–5947040 | 20/02/2019 18:23:27 | 20/02/2019 18:24 | <= 1 | 5 | 39.62 | 26.43 | 8 |
| 8-agustos-2019-izmir-depremi–6135297 | 08/08/2019 08:39:07 | 08/08/2019 08:39 | <= 1 | 4.6 | 38.02 | 26.85 | 6 |
| 20-agustos-2019-ankara-depremi–6148327 | 20/08/2019 02:07:35 | 20/08/2019 02:08 | <= 1 | 3.2(ml) | 39.89 | 33.04 | 10 |
| 25-agustos-2019-ankara-depremi–6155192 | 25/08/2019 18:42:26 | 25/08/2019 18:42 | <= 1 | 3.5(ml) | 40.04 | 32.8 | 5 |
| 24-eylul-2019-istanbul-depremi–6189374 | 24/09/2019 08:00:22 | 24/09/2019 08:00 | <= 1 | 4.5 | 40.88 | 28.21 | 4 |
| 26-eylul-2019-istanbul-depremi–6191375 | 26/09/2019 10:59:24 | 26/09/2019 11:00 | <= 1 | 5.6 | 40.88 | 28.21 | 5 |
| 10-ekim-2019-yalova-depremi–6208576 | 10/10/2019 16:52:03 | 10/10/2019 16:52 | <= 1 | 4 | 40.68 | 29.25 | 13.9 |
| 10-ekim-2019-yalova-depremi–6208576-2 | 10/10/2019 17:04:39 | 10/10/2019 17:04 | <= 1 | 3.1(ml) | 40.7 | 29.26 | 5 |
| 10-ekim-2019-yalova-depremi–6208576-3 | 10/10/2019 17:09:40 | 10/10/2019 17:10 | <= 1 | 3.3(ml) | 40.7 | 29.25 | 2 |
| 10-ekim-2019-yalova-depremi–6208576-4 | 10/10/2019 19:32:07 | 10/10/2019 19:32 | <= 1 | 3.7(ml) | 40.69 | 29.26 | 12 |
| 10-aralik-2019-balikesir-depremi–6277350 | 12/10/2019 20:14:02 | 12/10/2019 20:15 | <= 1 | 4.6 | 39.45 | 29.93 | 8 |
| 10-aralik-2019-balikesir-depremi–6277350-2 | 12/10/2019 20:24:05 | 12/10/2019 20:24 | <= 1 | 4.3 | 39.44 | 29.91 | 11.3 |
| 10-aralik-2019-balikesir-depremi–6277350-3 | 12/10/2019 20:46:18 | 12/10/2019 20:47 | <= 1 | 4 | 39.44 | 29.9 | 14.3 |



**Table 2.** Table of extracted information from entries of each topicID. Number of entries with city and district information along with the MCS value is represented by No of Entry. Unique number of cities and districts of these cities are given by No of City and No of District, respectively. Maximum MCS, minimum MCS and average MCS of these entries are represented by Max MCS, Min MCS and Av MCS, respectively. Entries with magnitude guesses regardless of location information are given in No of Mag.

| topicID | No of Entry | No of City | No of District | Max MCS | Min MCS | Av MCS | No of Mag |
|---|---|---|---|---|---|---|---|
| 2814519 | 78 | 11 | 37 | 5 | 2 | 2.65 | 2 |
| 3418720 | 38 | 4 | 28 | 4 | 1 | 2.18 | 1 |
| 3673799 | 73 | 6 | 33 | 4 | 2 | 2.30 | 3 |
| 4966334 | 51 | 2 | 23 | 5 | 2 | 2.53 | 4 |
| 5137587 | 85 | 5 | 21 | 4 | 1 | 2.68 | 6 |
| 5208242 | 26 | 4 | 15 | 3 | 1 | 2.12 | 6 |
| 5296414 | 20 | 6 | 14 | 4 | 1 | 2.10 | 0 |
| 5296414-2 | 20 | 6 | 14 | 5 | 1 | 2.40 | 0 |
| 5388936 | 151 | 12 | 60 | 6 | 1 | 2.26 | 9 |
| 5392358 | 3 | 1 | 3 | 2 | 2 | 2.00 | 1 |
| 5392358-2 | 46 | 7 | 22 | 5 | 1 | 1.98 | 9 |
| 5861331 | 42 | 4 | 23 | 4 | 2 | 2.48 | 3 |
| 5881852 | 53 | 4 | 27 | 4 | 1 | 2.21 | 2 |
| 5919561 | 63 | 3 | 18 | 4 | 1 | 2.37 | 13 |
| 5947040 | 62 | 7 | 38 | 4 | 1 | 1.90 | 5 |
| 6135297 | 101 | 6 | 32 | 5 | 1 | 2.17 | 16 |
| 6148327 | 30 | 1 | 6 | 4 | 1 | 2.87 | 7 |
| 6155192 | 129 | 1 | 9 | 4 | 1 | 2.27 | 5 |
| 6189374 | 152 | 5 | 41 | 5 | 1 | 2.59 | 9 |
| 6191375 | 233 | 12 | 57 | 6 | 1 | 2.74 | 1 |
| 6208576 | 68 | 3 | 18 | 5 | 1 | 2.15 | 15 |
| 6208576-2 | 12 | 3 | 10 | 3 | 1 | 2.00 | 0 |
| 6208576-3 | 25 | 2 | 12 | 2 | 1 | 1.96 | 0 |
| 6208576-4 | 29 | 3 | 10 | 3 | 1 | 2.10 | 2 |
| 6277350 | 47 | 8 | 26 | 3 | 1 | 1.94 | 10 |
| 6277350-2 | 19 | 7 | 13 | 5 | 1 | 2.16 | 1 |
| 6277350-3 | 10 | 3 | 8 | 5 | 1 | 2.20 | 1 |





**Table 3.** Table of PGA and PGV value intervals for calculation of intensity (I); values are taken from (Cataldi et al., in prep.).

| I | 1 | 2 | 3 | 4 | 5 | 6 | 7 | 8 | 9 | 10 |
|---|---|---|---|---|---|---|---|---|---|---|
| $PGA_{min}$(cm/$s^2$) | | 0.32 | 1.91 | 6.31 | 17.78 | 52.48 | 85.11 | 141.25 | 269.15 | 575.44 |
| $PGA_{max}$(cm/$s^2$) | $< 0.32$ | 1.91 | 6.31 | 17.78 | 52.48 | 85.11 | 141.25 | 269.15 | 575.44 | 1148.15 |
| $PGV_{min}$(cm/s) | | 0.01 | 0.10 | 0.28 | 0.74 | 2.57 | 5.75 | 9.77 | 21.38 | 39.81 |
| $PGV_{max}$(cm/s) | $< 0.01$ | 0.10 | 0.28 | 0.74 | 2.57 | 5.75 | 9.77 | 21.38 | 39.81 | 70.789 |