# Peer review of "Web-based macroseismic intensity study in Turkey: entries in Ekşi Sözlük"

_Geoscience Communication, 2020_

## Short Comment (SC1) · 17 Jul 2020

Ekşi Sözlük actually is a "collaborative hypertext dictionary" (a.k.a. "interactive dictionary" or "participatory dictionary" among Turkish people). And CHDs are commonly confused with web forums or other some things similar. This is usually because we introduce Turkish CHDs to foreigners and constantly compare them with other websites.

I actually did an introduction about this and Japanese web analytics consultant Ejiri Toshiaki shared it for me.

https://ejtter.com/130520205062/

In addition to this, I want to tell about differences between forums and CHDs:

[Types of posts] Forums: In any format, mostly as Q&A CHDs: Informations as "definitions" (and users' opinions if they want)

[Mentions as replies] Forums: YES CHDs: NO (Because CHD structure is not created as Q&A or another format which doesn't defining anything. On CHDs, the heroes of discussions are informations, not people. Entries usually aims to provide information.)

[Hiding some contents (links, images etc.) in the posts optionally] Forums: YES (may not be available in some forums) CHDs: NO (Because it's important for everyone to have access to all the necessary information. CHDs meet this need as well.)

---

## Short Comment (SC2) · 23 Jul 2020

I agree with you for the description of the eksisozluk. In fact, I describe it as "collaborative dictionary" in line 23. I will make the proper correction for its description in line 2 in which I describe it as "forum-like".

---

## Referee Comment (RC1) · Anonymous Referee #1 · 24 Aug 2020

The authors analyze the user entries for some earthquakes occurred in Turkey area; the related intensities are estimated and the results between the user entries and the shakemaps are compared. The results are very interesting and the conclusions address an issue of importance to microseismic intensities and shaking estimations; the paper is well written and structured and the topic well fits the aims and scope of the journal. For these reasons I recommend the publication.

---

## Referee Comment (RC2) · Anonymous Referee #2 · 30 Oct 2020

The manuscript touches on an interesting topic, by analysing the users' comments on earthquakes on a public, general purpose website in Turkey. Based on the comments, the authors predict the intensities using the ground motion prediction equations developed for Italy. They analyse four earthquakes, and they even attempt to crowd-source the magnitude estimations for the earthquakes. Unsurprisingly, these show the largest scatter.

The manuscripts uses a lot of unnecessary abbreviations. The description of the methodology is rather thin, the reader should expect a more detailed explanation what is exactly done. This section needs to be extended.

In the discussion the authors correctly point out that because of users' the location is known only at the district level, and taking the centroid location of the district introduces

errors due to the variability of soil conditions within the district. Moreover, changes in population density are only rudimentarily taken into account.

The manuscript makes an interesting observation: people who felt larger number of earthquakes, make better a guess about the magnitude of the earthquake than those who live in an aseismic region, although I'm not sure how does this helps in the determination of magnitude.

The authors observe that the residuals between observed and predicted MCS values increase with distance, but do not give any explanation why.

The future work section is a wish list for possible tasks for the public website, and therefore it is quite irrelevant for the paper itself. I recommend deleting it.

The only difference between the intensity maps and felt reports is that the locations of the felt reports are plotted on top of the intensity maps; there is no need for two separate figures.

I would recommend the acceptance of the manuscript with major revisions.

---

## Author Comment (AC1) · 2 Nov 2020

We are thankful to the anonymous reviewer for the encouraging comments. We are also glad to hear an acceptance of our study from the anonymous referee.

---

## Author Comment (AC2) · 10 Nov 2020

We are thankful for the improving comments of the Anonymous Referee #2. We have agreements and disagreements with some of those comments. We prove our updates and opinions about the referee's comments.

1. The manuscripts uses a lot of unnecessary abbreviations - There is only one abbreviation that is used once (DBMI15 in line 55). We delete that abbreviation.

2. The description of the methodology is rather thin, the reader should expect a more detailed explanation what is exactly done - We expand the Method section.

3. The manuscript makes an interesting observation: people who felt larger number of earthquakes, make better a guess about the magnitude of the earthquake than

those who live in an aseismic region, although I'm not sure how does this helps in the determination of magnitude - We compare the magnitude estimations of users that are living in regions where earthquakes with various magnitudes are more common with users that are living in regions where earthquakes are relatively rare. Our proposal on the user estimation relies on the fact that users who have experienced seismic activities more often have a baseline to compare their latest feeling with their previous experiences. On the other hand, users who live in less seismically active regions have a lack of such baseline. Hence they may think that they have felt the earthquake due to its larger magnitude. We explain it in the conclusion.

4. The authors observe that the residuals between observed and predicted MCS values increase with distance, but do not give any explanation why - We believe that it is due to the lack of explanation of users' experience. We express our thoughts in the updated version as below: "This is due to the fact that it is hard to distinguish between MCS 1 to MCS 2 by analyzing the entries. Lack of resolution in terms of 'expression of the experience' limits our distinction levels for intensity."

5. The future work section is a wish list for possible tasks for the public website, and therefore it is quite irrelevant for the paper itself. I recommend deleting it - We believe that in the case of an official collaboration with the website, it would be a nice data provider. Hence, we would like to point out the significance of a possible collaboration.

6. The only difference between the intensity maps and felt reports is that the locations of the felt reports are plotted on top of the intensity maps; there is no need for two separate figures - We agree with the referee and keep the figures with EMSC information and delete the other type of presentation.

---

## Author Response (AR1)

SC1: 'About a false fact', Mitsuki Yadate, 17 Jul 2020
1. Eksi Sözlük actually is a "collaborative hypertext dictionary" (a.k.a. "interactive dictionary" or "participatory dictionary" among Turkish people). - I agree with you for the description of the eksisozluk. In fact, I describe it as "collaborative dictionary" in line 23. I will make the proper correction for its description in line 2 in which I describe it as "forum-like".

RC1: 'review', Anonymous Referee #1, 24 Aug 2020
1. The authors analyze the user entries for some earthquakes occurred in Turkey area; the related intensities are estimated and the results between the user entries and the shakemaps are compared. The results are very interesting and the conclusions address an issue of importance to microseismic intensities and shaking estimations; the paper is well written and structured and the topic well fits the aims and scope of the journal. For these reasons I recommend the publication - We are thankful to the anonymous reviewer for the encouraging comments. We are also glad to hear an acceptance of our study from the anonymous referee.

RC2: 'Reviewer's comments', Anonymous Referee #2, 30 Oct 2020
1. The manuscripts uses a lot of unnecessary abbreviations - There is only one abbreviation that is used once. We delete these abbreviations in line 69.
2. The description of the methodology is rather thin, the reader should expect a more detailed explanation what is exactly done. - We expand the Method section.
3. In the discussion the authors correctly point out that because of users' the location is known only at the district level, and taking the centroid location of the district introduces errors due to the variability of soil conditions within the district. Moreover, changes in population density are only rudimentarily taken into account. - This is correct. Even if we implement the population density on top of the data that we collected, we do not have the neighborhood level information of these data.
4. The manuscript makes an interesting observation: people who felt larger number of earthquakes, make better a guess about the magnitude of the earthquake than those who live in an aseismic region, although I'm not sure how does this helps in the determination of magnitude. - We compare the magnitude estimations of users that are living regions where earthquakes with various magnitudes are more common with  users that are living regions where earthquakes are relatively rare. Our proposal on the user estimation relies on the fact that users experienced seismic activities more often have a baseline to compare their latest feeling with their previous experiences. On the other hand, users lives in less seismically active regions have lack of such baseline. Hence they may think that they felt the earthquake due to its larger magnitude. We explain this on lines between 205 and 209.
5. The authors observe that the residuals between observed and predicted MCS values increase with distance, but do not give any explanation why. - We believe that it is due to the fact of lack of explanation of users' experience. We express our thoughts in lines between 180 and 181.
6. The future work section is a wish list for possible tasks for the public website, and therefore it is quite irrelevant for the paper itself. I recommend deleting it. - We believe that in the case of a collaboration with the website, it'd be a nice data provider. Hence, we'd like to point out the significance of possible collaboration.
7. The only difference between the intensity maps and felt reports is that the locations of the felt reports are plotted on top of the intensity maps; there is no need for two separate figures. - We keep the figures with EMS information and delete the other type of presentation.

---

## Author Response (AR2)

Comments to the Author:
Many thanks for your revised version. I have some additional comments, mainly to strongly encourage you to remember that this is a non-specialist journal and has a broad readership across the geosciences. Many readers will not be familiar with some of the earthquake terminology used and so more explanation of many technical terms is necessary.

1) Abstract line 3 - shape or shake?
We change it as shake.

2) line 47 - what is the MCS scale? Many readers of this journal will not be familiar with this.
We explained the MSC scale in the same line.

3) line 49 - same as above for 'S waves'. Please remember that this journal is about geoscience communication and has a broad readership across disciplines.
We provide a explanation of S waves in terms of its significance in the context.

4) line 66 - you have deleted the definitions of these terms (PGV and PGA) used.
We explain PGA and PGV in the last paragraph of the Method section even though they are mentioned in the second paragraph of the same section. We make the proper explanation in the first usage of the parameters.

5) Section 3 - given the readership, please provide a simpler explanation of the methods, as well as the technical details.
We provide simpler explanations of various technical details.

6) line 86 - please provide the URL of the repository in the main text.
We are providing the URL in Code and data availability section. In order to avoid a repeated information, we deleted this sentence.

---

## Author Response (AR3)

Comments to the Author:

1) there are many uses of MCS and many of MSC throughout the paper - please ensure consistency.

We are deeply sorry for the inconsistent abbreviation. Correct abbreviation is MCS and we changed MSC to MCS.

2) the description of S waves and P waves still does not explain what they are to a broad audience. Please describe what is meant assuming a reader has no technical knowledge of earthquakes.

We deleted the P and S wave information and give an information which does not require a knowledge of earthquakes. Instead of giving our idea about the correlation between the creation time of the topics and the arrival of the seismic waves, now, we are providing a more precise information about the time gap between earthquake origin and topic creation time. We had, and still have, an opinion of a correlation between these two times. However, we cannot prove it due to the lack of seconds in the times provided by the website.

3) line 49 - 'can be change' -> 'can be different'

Done.